# The Association between Prosocial Behaviour and Peer Relationships with Comorbid Externalizing Disorders and Quality of Life in Treatment-Naïve Children and Adolescents with Attention Deficit Hyperactivity Disorder

**DOI:** 10.3390/brainsci11040475

**Published:** 2021-04-09

**Authors:** Szabina Velő, Ágnes Keresztény, Gyöngyvér Ferenczi-Dallos, Luca Pump, Katalin Móra, Judit Balázs

**Affiliations:** 1Doctoral School of Psychology, Eötvös Loránd University, 1064 Budapest, Hungary; pump.luca@ppk.elte.hu; 2Department of Developmental and Clinical Child Psychology, Institute of Psychology, Eötvös Loránd University, 1064 Budapest, Hungary; kereszteny.agnes@ppk.elte.hu (Á.K.); dallos.gyongyver@ppk.elte.hu (G.F.-D.); katalinmora.kk@gmail.com (K.M.); balazs.judit@ppk.elte.hu (J.B.); 3Department of Psychology, Bjørknes University College, 0456 Oslo, Norway

**Keywords:** prosocial behaviour, peer relationships, quality of life, ADHD, comorbidity, externalizing disorders, conduct disorder

## Abstract

Several recent studies confirmed that Attention Deficit Hyperactivity Disorder (ADHD) has a negative influence on peer relationship and quality of life in children. The aim of the current study is to investigate the association between prosocial behaviour, peer relationships and quality of life in treatment naïve ADHD samples. The samples included 79 children with ADHD (64 boys and 15 girls, mean age = 10.24 years, SD = 2.51) and 54 healthy control children (30 boys and 23 girls, mean age = 9.66 years, SD = 1.73). Measurements included: The “Mini International Neuropsychiatric Interview Kid; Strengths and Difficulties Questionnaire” and the “Inventar zur Erfassung der Lebensqualität bei Kindern und Jugendlichen”. The ADHD group showed significantly lower levels of prosocial behaviour and more problems with peer relationships than the control group. Prosocial behaviour has a weak positive correlation with the rating of the child’s quality of life by the parents, both in the ADHD group and in the control group. The rating of quality of life and peer relationship problems by the parents also showed a significant negative moderate association in both groups. The rating of quality of life by the child showed a significant negative weak relationship with peer relationships in the ADHD group, but no significant relationship was found in the control group. Children with ADHD and comorbid externalizing disorders showed more problems in peer relationships than ADHD without comorbid externalizing disorders. Based on these results, we conclude that therapy for ADHD focused on improvement of prosocial behaviour and peer relationships as well as comorbid externalizing disorders could have a favourable effect on the quality of life of these children.

## 1. Introduction

Prosocial behaviour does not have a generally accepted, unified definition, but researchers tend to agree to use it as an umbrella term for several behaviours, including helper, supportive, sharing, cooperative and politeness behaviour, without the expectation of a possible reward [1,2]. The first appearance of these behaviours is at around 2 years of age; emotional atonement and empathy play an important role in their development [3]. Previous studies found that with age and the development of selfhood, prosocial behaviour also develops through the experience of social interactions [4]. Prosocial behaviour contributes to harmonic relationships in the family, positive social relationships and friendships [5,6,7]. Primary school children, who perform high in measures of prosocial behaviour, perceive acceptance and positive social relationships from their peers [5]. 

Attention-Deficit/Hyperactivity Disorder (ADHD) is one of most common neurodevelopmental disorders, affecting 4–6% of the primary school population [8,9]. Its occurrence is more common among boys: the gender distribution proportion is 3:1 [10]. The core symptoms of ADHD are poor attentional performance, impulsivity and hyperactivity [11,12]. According to the latest, fifth edition of the Diagnostic and Statistical Manual of Mental Disorders (DSM-5) at least six (five above age of 17) of the nine symptoms of attention deficit and/or at least six (five above age of 17) of the nine symptoms of hyperactivity must be present to fulfil diagnosis of ADHD. Additional criteria include the onset of symptoms before the age of 12 years, the persistence of at least 6 months, and the impairment of function in at least two situations [11]. Based on previous studies, two-thirds of children diagnosed with ADHD are diagnosed with at least one comorbid psychiatric disorder [13,14,15]. Conduct disorder (CD) is the most common comorbid disorder, appearing in 20% of cases of ADHD [16,17]; moreover, oppositional defiant disorder (ODD) was also found to be one of the most common comorbid diseases [18,19]. According to the DSM-5, conduct disorder (CD) is a recurrent and persistent pattern of behavior in which a child or adolescent violates the fundamental rights of others or more important social norms and rules appropriate to age, while oppositional defiant disorder (ODD) manifests itself as a rebellion against an authority [11].

ADHD is associated with cognitive, social and emotional impairments [20,21] and negatively affects the child’s relationship with family members [22,23,24]. Paap et al. [25] found a relationship between peer problems and prosocial behaviour in typically developing 7–9 years old children, but ADHD and ODD act as moderator variables, weakening this relationship. Tengsijaritkul and their colleagues [26] examined functional impairments in treated ADHD children and found that they have lower prosocial scores; moreover, their comorbid medical disorders were associated with higher problem scores. Furthermore, in a clinical study, peer problems and prosocial behaviour of methylphenidate treated children with ADHD and children without ADHD were compared [27]. The results indicated that children with ADHD show more problems with peers than children without ADHD and teachers appreciated them less as prosocial [27]. Comorbidity—specifically externalizing comorbid disorders—contributes to increase social difficulties among children with ADHD [28]. 

Measuring quality of life could be important for the investigation of function impairment and therapeutic effectiveness among childhood psychiatric disorders, including ADHD [29,30,31,32]. Functional impairment is a criterion for all psychiatric disorders according to the classification systems. Furthermore, in the case of ADHD, functional impairment needs to be present in at least in two areas, i.e., school/work and social life [11,12]. The concept of quality of life is a multidimensional measure that is broader than functional impairment, as it encompasses the overall health, impairments and effectiveness in several areas of daily life, including academic settings, leisure activities, and social life with family and friends [33]. All these areas can be affected by the presence of a mental disorder such as ADHD; thus, the assessment of quality of life could add valuable information about the patient’s current status in regard to the focus of treatment as well as measuring its efficacy. In the light of the above-mentioned findings, in the last decade several researchers investigated the effect of ADHD on quality of life [30,31]. Previous studies confirmed that ADHD has a negative influence on the child’s quality of life; these children have a lower level of quality of life than their healthy peers [30,31]. Effective multimodal treatment exists for the management of ADHD, including parental education, cognitive behavioural therapy and medication [34,35,36,37]. Clinical studies have indicated that pharmacotherapy/multimodal treatment has a positive effect on quality of life and on the remission of ADHD symptoms [29,32].

Peer relationships are important for the social development of children [38]. Children diagnosed with ADHD often have difficulties developing peer relationships due to their ADHD symptoms, such as impulsivity and poor attention [39]. Therefore, it is important to examine possible factors connecting to their social functioning, such as their prosocial behavior. Furthermore, children with ADHD have significantly lower quality of life compared to healthy children in many areas including peer relations [40]. Previous studies have indicated that quality of life can be an important tool to measure the impact of a mental disorder and for assessing the effectiveness of a treatment [29]. 

Based on these findings, we examined if a better understanding of peer relationship and prosocial behavior can improve the quality of life of children with ADHD. Additionally, the assessment of prosocial skills and prosocial behavior in treatment naïve children with ADHD can serve as a baseline measurement for monitoring the efficacy of therapies. Although several previous studies have examined the functional impairment in children with ADHD, including such aspects as social functions, peer functioning and prosocial behaviour, the effectiveness of treatment in clinical trials is not filtered out, which is important when we want to evaluate functional impairment [26,27,28]. To our knowledge, there has not been any research conducted which explored prosocial behaviour and peer relationships among treatment naïve children with ADHD, nor are we aware of any research that explored the relationship between prosocial behaviour, peer relationships and quality of life.

The aim of the current study was to investigate the levels of prosocial behaviour and peer relationship problems among samples which were carefully selected: 1. a treatment-naïve ADHD group of children who were diagnosed both by a child psychiatrist and a structured diagnostic interview, and 2. a control group of children with no previously recognized psychiatric disorders or any psychiatric disorders currently diagnosed by a structured diagnostic interview. Moreover, we wanted to investigate the relationship between prosocial behaviour, peer relationships and quality of life (both the parents’ and the child’s ratings) in both the control sample and the treatment naïve ADHD sample. Finally, our goal was to explore the differences, in terms of prosocial behaviour and peer relationships, between those in the treatment naïve ADHD sample who had comorbid externalising disorders (i.e., CD and/or ODD) and those who did not have comorbid externalising disorders.

**Hypothesis** **1.**
*Treatment naïve children with ADHD show a lower level of prosocial behaviour and a higher level of peer relationship problems than healthy children.*


**Hypothesis** **2.**
*A higher level of prosocial behaviour is associated with a higher level of quality of life in treatment naïve children with ADHD and healthy children based both on parental proxy- and children’s self-reports.*


**Hypothesis** **3.**
*A higher level of peer relationship problems is associated with a lower level of quality of life in the treatment naïve children with ADHD and healthy children based both on parental proxy and children’s self-report.*


**Hypothesis** **4.**
*A lower level of prosocial behaviour is associated with a higher level of peer relationship problems in treatment naïve children with ADHD and healthy children.*


**Hypothesis** **5.**
*Treatment naïve children diagnosed with ADHD and comorbid externalizing disorders show a lower level of prosocial behaviour and a higher level of peer relationship problems than treatment naïve children with ADHD and without comorbid externalizing disorders.*


## 2. Materials and Methods

### 2.1. Recruitment and Research Participants

We admitted into our study a treatment-naïve ADHD group and a healthy control group of children aged 6 to 18 years. The ADHD group of children was recruited from the Vadaskert Child and Adolescent Psychiatric Hospital and Outpatient Clinic, Budapest, Hungary. We used the following inclusion criteria for the treatment-naive ADHD group: children with a diagnosis of ADHD according to a structured diagnostic interview (see below), no previous psychological and/or psychiatric treatment including both psychotherapy and pharmacotherapy in the medical history. We enrolled these children into our study after their psychiatrists diagnosed them with ADHD in the hospital/out-patient clinic, but before their treatment started.

In the clinical group, the child was not included into the study if the child’s psychiatrist indicated that intellectual disability had been suspected or confirmed previously or during the current hospital examination. From the clinical group, two different groups were created: one with comorbid externalizing problems, and one without comorbid externalizing problems. Among the ADHD group with externalizing problems, besides the ADHD diagnoses, CD and/or ODD had to be present. Among the group without externalizing problems, neither of these two diagnoses were present.

To create the control group, twelve schools were randomly selected from a list of public primary schools of Budapest. Furthermore, two schools from countryside were included, which were selected by researchers. Only primary schools educating children with average intelligence were included. Special schools that educate children with mental retardation have been excluded. Children with any ongoing or previous psychological or psychiatric treatment were also excluded. The absence of any psychiatric disorders was confirmed by a structured psychiatric interview (see below).

### 2.2. Characteristics of Sample

The treatment naïve ADHD group consisted of 79 children: 64 (81%) boys and 15 (19%) girls. The mean age of children with ADHD was 10.24 years (SD = 2.51, range: 6–15). Among the ADHD group, 49 (62.8%) children were diagnosed with an externalizing comorbid disorder (CD and/or ODD), while 29 (37.2%) children were not diagnosed with any comorbid externalizing disorders. Gender distribution in the ADHD group without externalizing disorders: 25 (86.2%) boys, 4 (13.8%) girls; while, in the ADHD group with externalizing disorders: 38 (77.6%) boys, 11 (22.4%) girls. The gender and the group (ADHD with or without externalizing disorders) revealed no significant relationship (*χ*^2^(1) = 0.879, *p* = 0.349).

The control group consisted of 54 children: 31 (57.4%) boys and 23 (42.6%) girls. The mean age of children in the control group was 9.66 years (SD = 1.73, range: 6–14). There was a non-significant difference in age between the ADHD and the control group (U = 2267, *p* = 0.344). The gender and the group (ADHD-control) revealed a significant relationship (*χ*^2^(1) = 8.758, *p* = 0.003). The percentage of boys in the clinical group was 81% (64 boys); in the control group, it was 57.4% (31 boys). There was no gender and age difference in any of the variables examined (see Table 1 and Table 2).

### 2.3. Procedure

This study was approved by the Ethical Committee of the Medical Research Council, Hungary (ETT-TUKEB), project identification code: 26182/2011-EKU. The parents of each child and adolescent were provided written informed consent after being informed of the nature of the study. Children/adolescents participated in a diagnostic interview recorded by a psychologist and then completed questionnaires related to the study. No compensation was provided to the participants.

### 2.4. Measures

#### 2.4.1. Psychiatric Symptoms and Diagnoses

To measure psychopathology and diagnoses both in the clinical and healthy control groups, the modified version of the Hungarian Mini International Neuropsychiatric Interview for Children and Adolescents (MINI Kid) 2.0 [41,42,43,44] was applied. The MINI Kid is a structured psychiatric interview which assesses the 25 DSM-IV child and adolescent psychiatric disorders. The modified version of the MINI Kid evaluates not only psychiatric disorders, but assesses all psychiatric symptoms, enabling subthreshold disorders to be identified. The interview is suitable for children aged between 6 and 18 years; it was administered to children under 13 years of age in the presence of their parents, while those who were 13 years of age and above of age participated in the interview on their own.

#### 2.4.2. Prosocial Behaviour, Peer Problems

We used the Hungarian version of the Strengths and Difficulties Questionnaire (SDQ) [45,46], which serves to explore and filter childhood behavioural problems and mental disorders. The questionnaire consists of 25 items; each item can be scored from 0 to 2 (answer possibilities: 0 = not true, 1 = somewhat true, 2 = firmly true), and each scale point is ranged between 0 and 10. The items of the questionnaire are classified into 5 subscales: emotional symptoms, behavioural problems, hyperactivity, peer relationship problems, and prosocial behaviour. In the present study, we focused on the answers given to the prosocial behaviour and peer relationship problems subscales.

#### 2.4.3. Measuring Quality of Life

The Quality of Life Questionnaire [47], or “Inventar zur Erfassung der Lebensqualität bei Kindern und Jugendlichen” (ILK) [48], is a Hungarian version of a subjective measurement of life quality. The original questionnaire consists of 15 items, which pertain to school, family relationships, time spent with peers, time spent alone, and finally, physical and mental health [48]. The measurement is suitable for children from 7 to 18 years of age. It has a self-rated version both for children and adolescents, and a proxy parent rated version as well. The questionnaire measures in a 5-point Likert scale. Descriptive statistics and internal consistency of the measurements can be found in Table 3.

### 2.5. Statistical Analysis

After data recording, a 10% inspection followed by data cleaning was performed to create a valid database. Salient cases, i.e., cases where the response exceeded the minimum or maximum score of the questionnaire, have been excluded. The distribution of relative frequencies and the descriptive analysis with means and standard deviations were calculated to describe the sample characteristics and the used measurements; Cronbach’s alpha was used to measure the internal consistency of the measurements. The Shapiro–Wilk test was applied to test the normality of peer relationships problems, prosocial behaviour and quality of life (ILK). As none of the variables show normal distribution, non-parametrical tests were applied. In order to examine the differences in prosocial behaviour and peer relationship problems between the clinical group and the control group, and between the ADHD group with externalizing problems and the ADHD group without externalizing problems Mann–Whitney U tests were performed. Spearman rank correlation coefficients were calculated for evaluating monotonous associations between prosocial behaviour, peer relation problems and self- and parent-rated quality of life. Post hoc power analysis was calculated using G*Power software for detecting significant effects of the results analysed [49]. Statistical procedures were performed using IBM SPSS 25 statistical software [50].

## 3. Results

### 3.1. H1. Prosocial Behaviour and Peer Relationship Problems in the Treatment Naïve ADHD and Control Groups

The treatment naïve ADHD group showed a significantly lower value in prosocial behaviour than the control group (power × 0.77). The treatment naïve ADHD group’s value in peer relationship problems was significantly higher than that of the control group (power × 1.00) (see Table 4).

### 3.2. H2–H4. Prosocial Behaviour’s Association with Peer Relationships and Quality of Life

Table 5 presents the relationship between prosocial behaviour, peer relationship problems and quality of life.

Prosocial behaviour had a weak positive relationship with the parent’s evaluation of the child’s quality of life both in the ADHD and in the control group. The parent’s rating of quality of life and peer relationship problems also showed a significant, negative moderate association in the ADHD and in the control group. Between prosocial behaviour and peer relationship problems, a negative weak association was detected in the ADHD group, and a negative moderate association in the control group. Furthermore, in the ADHD group, the child’s evaluation of quality of life showed a significant negative weak relationship with peer relationships, and there was no significant relationship with prosocial behaviour. In the control group, the child’s view of life quality did not show a significant relationship with the other items.

### 3.3. H5. Comorbid Externalizing Problems, Prosocial Behaviour and Peer Relationship Problems in the Treatment Naïve ADHD Group

The two treatment naïve ADHD groups—i.e., (1) children diagnosed with externalizing comorbid disorders (CD and/or ODD); and (2) children not diagnosed with externalizing comorbid disorders—showed no significant difference in prosocial behaviour (power × 0.37). The two groups showed a significant difference in peer relationship problems. The ADHD group with externalizing comorbid disorders showed higher values in peer relationship problems than the ADHD group without externalizing disorders (power × 0.81) (see Table 6).

## 4. Discussion

To our knowledge, the current study was the first to investigate the association between prosocial behaviour and peer relationship problems and quality of life among treatment naïve children diagnosed with ADHD. Furthermore, the present study compared a carefully selected homogeneous treatment-naïve ADHD group with a control group of children with no previously recognized psychiatric disorders or any psychiatric disorders currently diagnosed by a structured diagnostic interview.

Based on our results, we can establish that treatment-naïve children with ADHD show lower levels of prosocial behaviour than healthy children; moreover, they have more problems with their peers. These results are consistent with the findings of Paap et al. [25], which showed high levels of ADHD symptoms and behavioural problems (as perceived by teachers and parents) were associated with low levels of prosocial behaviour and high levels of peer relationship problems. Because of their attentional difficulties, children diagnosed with ADHD are handicapped in those social skills which are learned through observation. In addition, hyperactive and impulsive behaviours contribute to generally unrestrained and overbearing social behaviour that makes children with ADHD highly aversive to peers [39]. Hay et al. [51] also suggest that there may be specific neurobiological deficits making it difficult for children with ADHD symptoms to regulate their attention and activity sufficiently to deploy prosocial behaviour.

The present study explored the association between prosocial behaviour, peer relationship problems and parents’ and child’s evaluation of life quality in a treatment naïve ADHD group and a control group. A notable finding of our research was that a weak positive relationship exists between prosocial behaviour and the parental evaluation of quality of life both in the treatment naïve ADHD group and the control group, while the self-reported quality of life did not reveal an association with prosocial behaviour in either group. This finding suggests that higher prosocial values have a positive correlation with the child’s quality of life, as evaluated by their parents, in both groups, while, in contrast, they do not have this positive correlation with quality of life evaluated by the child itself. Previous studies have also highlighted that there may be a difference in perception of quality of life between parents and their children [30,31]. When examining the quality of life of children, it has become increasingly clear that in addition to the child’s own subjective judgment, an external observer is also important. The use of proxy reports is recommended in order to obtain a more extensive and reliable picture of the situation of children and adolescents [52]. According to Mattejat et al. [48], the evaluation by a parent or caregiver is both subjective and objective because, although they appear as external observers, they are themselves affected by the child’s condition. According to Thaulow and Jozefiak [53], the parent–child difference in quality of life perceptions is due to children with ADHD being more likely to focus on the present aspects, while their parents are more likely to focus on the child’s future, which is concerned with school and social problems. Presumably, parents may have a greater insight into the child’s difficulties, so it is important when assessing the quality of life of children with ADHD to take both the parent’s and the child’s perspective into account. For instance, the perceived quality of life of a child that is prosocial and shares its toys may not always improve, as peers may not always reciprocate this kindness. In exploring the association between peer problems and quality of life, we found that the evaluation by the parent has a negative but moderate relationship with peer problems, in both the treatment naïve ADHD group and the control group. In addition, when parents report fewer peer relationship problems, they also rate their child’s quality of life more highly. The association between these variables is also detectable in the self-reports of treatment-naïve ADHD children, so we can see that as children with ADHD perceive more problems in their peer relationships, they rate their quality of life lower; however, this association is not detectable in the self-reports of the control group. Previous studies have confirmed that children with ADHD perceive more rejection from their peers then healthy children [39], which is likely to negatively affect their evaluation of their quality of life. Furthermore, there was a weak negative relationship between prosocial behaviour and peer relationship problems in the treatment naïve ADHD group, and a negative moderate relationship in the control group. This result reveals the importance of prosocial behaviour for peer relationships. The findings highlight that treatment naïve children with ADHD do experience an association between peer relationship problems and quality of life, but not between prosocial behaviour and quality of life, while parents experience both associations. It is important to note that effective therapy for ADHD should not only relieve ADHD symptoms but also improve the child’s quality of life [29]. As effective treatments geared at other aspects of dysfunction associated with ADHD do not eradicate ADHD children’s peer problems, peer problems need to be targeted directly [39]. Lowering the number of peer relationship problems could have a favourable effect on quality of life. Preventive and interventional programmes, focusing on the easement of peer relationship problems, could help reduce experiences of exclusion and enhance quality of life among children with ADHD.

Since most children with ADHD can be diagnosed with a comorbid psychiatric disorder [13,14,15], the present study wanted to explore whether the externalizing comorbid disorders affect prosocial behaviour and peer relationship problems. As we stated earlier, the results of the present study show that children with ADHD show lower levels of prosocial behaviour than the healthy controls; however, these differences are not detectable when we focus on the children with ADHD diagnoses with or without externalizing comorbid diagnoses. Hay et al. [51] found that aggressive behavioural symptoms were not associated with prosocial behaviour when they took ADHD symptoms into account. We must mention that the statistical power was low in testing this hypothesis due to a relatively small sample size. Whereas there was not a detectable difference in prosocial behaviour between the two groups, children with comorbid externalizing problems are characterised as having more peer relationship problems. CD or ODD can contribute to the child’s difficulties with peer relationships. Considering that there is an association between peer relationship problems and quality of life, therapy that focuses on comorbid externalizing disorders could contribute to reducing peer relationship problems and thus enhance the quality of life of children with ADHD.

The findings of the current study must be interpreted in light of certain limitations. First, our study was cross-sectional, which does not allow for any causational conclusions. Based on our results, we can, however, state that we found significant differences between healthy children and treatment naïve children with ADHD in the studied variables, which highlights the importance of learning more about prosociality and peer relationships through further research. Second, there was a difference in gender distribution between the clinical and control groups. This difference, however, reflects the general difference in gender ratio between children with ADHD, especially in clinical practice [54], and healthy children [55]. Additionally, it is important to note that the results of the current study were not affected by gender. Third, we did not use any structured measurement of intelligence test to assess the mental ability of children included in our study to reduce the study-load for the included children. Instead, children were encouraged to indicate whether they could understand the questions at the start of the study. Furthermore, each child was accompanied by a study mentor (i.e., parent or researcher) during the completion of the self-rating questionnaire, which made it possible for children to ask for information if needed. Fourth, it is considered a limitation of the study that the MINI Kid diagnostic interview applied in the study was still based on DSM-IV criteria instead of DSM-5. The reason for this was that the DSM-5-based version of the MINI Kid was not yet available at the start of the present study. However, we believe that the differences between the two versions are not essential in the case of the diagnoses of ADHD in children [56]. While the number of necessary ADHD symptoms above age 17 has changed from six to five and the onset of symptoms and impairments changed from 7 to 12, according to recent studies these changes have not affected childhood prevalence [57]. Fifth, the comparison of prosocial behaviour in children with ADHD and ADHD with comorbid externalizing disorders would need to be tested on a larger sample since, in this case, the power value was 0.38. For the other hypotheses, the statistical power was adequate (H1/1: power × 0.77; H1/2: power × 1.00; H5/2: power × 0.81).

The results of the present study indicate that peer relationship problems, prosocial behaviour, and their relationship to quality of life of children with ADHD are important areas for future research, preferably in a longitudinal design. Understanding which factors play a role in prosociality and peer relationships in children diagnosed with ADHD, could provide valuable insights in the development of ADHD symptoms, as well as their quality of life.

## 5. Conclusions

In summary, our research points out that treatment-naïve children diagnosed with ADHD have a lower level of prosocial behaviour and have more peer relationship problems than children not diagnosed with ADHD. Focusing on prosocial behaviour in ADHD therapy could have a favourable effect both on peer relationship problems and on quality of life, since prosocial behaviour has a positive relationship with quality of life, while peer relationship problems have a negative relationship with quality of life. Moreover, based on our study, therapy focusing on comorbid externalizing diagnoses could contribute to reducing peer relationship problems, and enhance quality of life. Social support from parents, teachers, as well as peers is important for all patients with psychiatric disorders, including ADHD. Psychoeducation can be an important factor for parents and teachers to improve acceptance and support towards children with ADHD. However, peers cannot be expected to be more accepting and tolerant to children with ADHD as a result of psycho-education, because they are themselves children as well. Therefore, children diagnosed with ADHD may need extra support to improve their relationship with their peers. Reducing social exclusion and improving peer relationships in addition to effective medication and non-medication therapies can help to protect children diagnosed with ADHD from future loneliness or deviations.

## Figures and Tables

**Table 1 brainsci-11-00475-t001:** Gender differences in parent-and child related quality of life, prosocial behaviour and peer relationship problems in the Attention-Deficit/Hyperactivity Disorder (ADHD) and control group.

	ADHD (*n* = 79)	Control (*n* = 54)
Gender distribution	64 (81%) boys; 15 (19%) girls	31 (57.4%) boys; 23 (42.6%) girls
Parent related quality of life	U = 459.50; *p* = 0.819	U = 339.00; *p* = 0.970
Child related quality of life	U = 396.50; *p* = 0.423	U = 307.50; *p* = 0.961
Prosocial behaviour	U = 457.50; *p* = 0.602	U = 385.50; *p* = 0.164
Peer relationship problems	U = 372.00; *p* = 0.694	U = 268.00; *p* = 0.206

**Table 2 brainsci-11-00475-t002:** Correlation between age and peer relationship problems and prosocial behaviour.

	Peer Relationship Problems	Prosocial Behaviour
	rho, *p*	rho, *p*
Age	0.095; 0.296	−0.132; 0.143

**Table 3 brainsci-11-00475-t003:** Descriptive statistics and internal consistency of the measurements in the present sample.

	Range	M (SD)	Median	IQR	Cronbach’s α
ILK—Judgement by parents	12–35	28.12 (5.40)	28	10	0.911
ILK—Judgement by children	17–35	29.17 (3.85)	30	6	0.682
SDQ—Peer relationship problems	0–9	2.47 (2.43)	2	4.50	0.788
SDQ—Prosocial behaviour	0–10	7.43 (2.29)	8	3.50	0.814

Note. M = Mean; SD = Standard deviation; IQR = interquartile range.

**Table 4 brainsci-11-00475-t004:** Difference between ADHD and control group in prosocial behaviour and peer relationship problems.

	ADHD Group (*n* = 79)			Control Group (*n* = 54)			
	**M (SD)**	**Median**	**IQR**	**M (SD)**	**Median**	**IQR**	**Mann–Whitney U test**
Prosocial behaviour	6.66 (2.40)	7	4	8.31 (1.84)	9	3	U = 1114.50; *p* < 0.0001
Peer relationship problems	4.05 (2.47)	4	4	0.73 (0.90)	0	1	U = 3244.00; *p* < 0.0001

Note. ADHD= Attention Deficit Hyperactivity Disorder; M = Mean; SD = Standard deviation; IQR = interquartile range.

**Table 5 brainsci-11-00475-t005:** Correlation between QoL and both peer problems and prosocial behaviour in the control and the ADHD groups.

	Peer Relationship Problems		Prosocial Behaviour	
Group	ADHD	Control	ADHD	Control
	rho; *p*		rho; *p*	
Parent-related quality of life	−0.658 **; 0.000	−0.470 **; 0.000	0.341 **; 0.004	0.374 **; 0.007
Self-related quality of life	−0.349 **; 0.003	−0.132; 0.361	0.024; 0.840	0.202; 0.163
Peer relationship problems		-	−0.289 *; 0.016	−0.439 **; 0.001

Note. QoL = Quality of life; ADHD = Attention Deficit Hyperactivity Disorder; ** The correlation is significant at the 0.01 level; * the correlation is significant at the 0.05 level.

**Table 6 brainsci-11-00475-t006:** Prosocial behaviour and peer relationship problems among children diagnosed with ADHD and among children diagnosed with ADHD and externalizing comorbid diagnoses.

	ADHD Group (*n* = 29)			ADHD + CD and/or ODD Group (*n* = 49)			
	**M (SD)**	**Median**	**IQR**	**M (SD)**	**Median**	**IQR**	**Mann–Whitney U test**
Prosocial behaviour	7.14 (2.28)	7	4.75	6.38 (2.45)	7	3	U = 539.00; *p* = 0.279
Peer relationship problems	3.08 (2.59)	2	4	4.58 (2.25)	4.5	3	U = 771.50; *p* = 0.017

Note. M = Mean; SD = Standard deviation; IQR = interquartile range; ADHD= Attention Deficit Hyperactivity Disorder; CD=Conduct Disorder; ODD= Oppositional Defiant Disorder.

## Data Availability

The data presented in this study are available on request from the corresponding author. The data are not publicly available due to privacy/ethical restrictions.

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
