# Peer review of "The Association between Prosocial Behaviour and Peer Relationships with Comorbid Externalizing Disorders and Quality of Life in Treatment-Naïve Children and Adolescents with Attention Deficit Hyperactivity Disorder"

_brainsci, 2021, doi:10.3390/brainsci11040475_

Round 1
Reviewer 1 Report
The major contribution of this research is that it examined these relationships in a group of children diagnosed with ADHD who had not undergone any treatment. However, that aspect is not present in the title of the paper and the authors did not provide enough information in the Introduction for the reader to understand why it is important to examine these questions in such a population. The introduction also lacked a strong transition when the authors began discussing quality of life--there was no indication of how this construct related to the social impairments that were discussed in the prior paragraph or even why it would be important to examine that relationship. There has to be a stronger rationale for the study than simply that no other researchers have done it.
In terms of the Method, I was surprised that the authors indicated that mental retardation, which is now called intellectual disability, was an exclusion criterion, but they didn't actually include any simple screening measure of intellectual functioning. They mentioned in the discussion that it was a limitation to rely on psychiatrist report of any previous indication of intellectual disability for the ADHD group and to rely on recruiting the control group only from public schools that serve "children with average intelligence." However, without an actual measure of intellectual functioning they really cannot say that intellectual disability was an exclusion criterion. I was also surprised that the diagnostic interview they used was based on DSM-IV criteria for mental disorders when the DSM-5 was published 8 years ago. Psychometric properties for the measures in the Method section were also not described accurately at times (e.g., confusing validity and reliability) or adequately.
The Results were well organized and generally clear (with the exception of typos and English language usage errors). However, no power analysis was presented to determine if it even made sense to conduct analyses comparing the ADHD only group with the ADHD + comorbid externalizing disorders group, given the small sample sizes. Again, sample size was mentioned as a limitation, but a lack of power was not adequately discussed as a possible explanation for the findings.
The Discussion was well organized and provided the reader with some good information about why the study was important in terms of using the research to identify potential targets for intervention. However, I thought the researchers could have provided more of an explanation for the different pattern of relationships when examining parent-reported QoL compared to child-reported QoL. I also thought the authors could have explained the limitations of the study more including how those limitations could have impacted their results, rather than just listing the limitations. Lastly, there was no discussion of future research directions.
In terms of the writing, I generally appreciated the concise nature of the article. However, in addition to the typos and English language usage errors there were several instances of causal language (e.g., "effect" and "influence") being used to describe correlational findings, which should be changed.
Author Response
We would like to thank the reviewer for the valuable feedback, comments and suggestions. Please find our responses below. We have modified the manuscript according to the feedback of the reviewer, unless otherwise specified below. All modifications are highlighted in the revised manuscript.
“Q1. The major contribution of this research is that it examined these relationships in a group of children diagnosed with ADHD who had not undergone any treatment. However, that aspect is not present in the title of the paper and the authors did not provide enough information in the Introduction for the reader to understand why it is important to examine these questions in such a population.”
We would like to thank the reviewer for this comment. In response we have now revised the title:
Line 2: “The Association between Prosocial Behaviour and Peer Relationships with Comorbid Externalizing Disorders and Quality of Life in Treatment-Naïve Children and Adolescents with Attention Deficit Hyperactivity Disorder”
Furthermore, in the revised manuscript we have now added the following paragraph to the Introduction section about the importance of our research questions:
Line 122: “Peer relationship is important for social development of children (38). Children diagnosed with ADHD often have difficulties developing peer relationships due to their ADHD symptoms, such as impulsivity and poor attention (39). Therefore, it is important to examine possible factors connecting to their social functioning, such as their prosocial behavior. Furthermore children with ADHD have significantly lower quality of life compared to healthy children in many areas including peer relations (40). Previous studies have indicated that quality of life can be an important tool to measure the impact of a mental disorder and for assessing the effectiveness of a treatment (29).
Line 129: “Based on these findings, we examined if a better understanding of peer relationship and prosocial behavior can improve the quality of life of children with ADHD. Additionally, the assessment of prosocial skills and prosocial behavior in treatment naïve children with ADHD can serve as a baseline measurement for monitoring the efficacy of therapies.”
“Q2.The introduction also lacked a strong transition when the authors began discussing quality of life--there was no indication of how this construct related to the social impairments that were discussed in the prior paragraph or even why it would be important to examine that relationship. There has to be a stronger rationale for the study than simply that no other researchers have done it.”
We would like to thank the reviewer for this comment. In response to this comment we have added the following paragraph in the revised manuscript:
Line 93: “Functional impairment is a criterion for all psychiatric disorders according to the classification systems. Furthermore, in the case of ADHD, functional impairment needs to be present in at least in two areas, i.e. school/work, social life (11,12). The concept of quality of life is a multidimensional measure that is broader than functional impairment, as it encompasses the overall health, impairments and effectiveness in several areas of daily life, including academic settings, leisure activities, and social life with family and friends (33). All these areas can be affected by the presence of a mental disorder such as ADHD, thus the assessment of quality of life could add valuable information about the patient's current status in regard to the focus of treatment as well as measuring its efficacy.”
Line 129: “Peer relationship is important for social development of children (38). Children diagnosed with ADHD often have difficulties developing peer relationships due to their ADHD symptoms, such as impulsivity and poor attention (39). Therefore, it is important to examine possible factors connecting to their social functioning, such as their prosocial behavior. Furthermore children with ADHD have significantly lower quality of life compared to healthy children in many areas including peer relations (40). Previous studies have indicated that quality of life can be an important tool to measure the impact of a mental disorder and for assessing the effectiveness of a treatment (29).”
“Q3. In terms of the Method, I was surprised that the authors indicated that mental retardation, which is now called intellectual disability, was an exclusion criterion, but they didn't actually include any simple screening measure of intellectual functioning. They mentioned in the discussion that it was a limitation to rely on psychiatrist report of any previous indication of intellectual disability for the ADHD group and to rely on recruiting the control group only from public schools that serve "children with average intelligence." However, without an actual measure of intellectual functioning they really cannot say that intellectual disability was an exclusion criterion.”
We thank the reviewer's for the comment regarding ‘mental retardation’, we have now replaced this term with ‘intellectual functioning’ in the revised manuscript.
Furthermore, please let us share why we did not use an intelligence test in regard to our exclusion criteria in the study.
The evaluation of the instruments used in the study (MINI Kid, SDQ, ILK) takes about 60-90 minutes. Due to our experience, a longer examination would have been impossible in one occasion for the children with ADHD, due to the nature of their ADHD symptoms. The study of the clinical group was performed at the Vadaskert Child Psychiatric Hospital and Outpatient Clinic, Budapest, which receives patients from all over the country, so it would not have been un-realistic to ask them to travel so much two times for a voluntary research.
Intellectual disability was an exclusion criterion, because it was important that the participants can understand the interview questions, however IQ was not a covariate in the study, therefore the aim of this exclusion criteria was not to ensure the homogenity of the study group. Therefore we found it sufficient to rely on the history and the reports of the child’s psychiatrist about his/her intellectual abilities.
Based on the reviewer’s comment intellectual disability was deleted from the exclusion criteria
Line 226: “In the clinical group the child was not included into the study, if the child's psychiatrist indicated that intellectual disability had been suspected or confirmed previously or during the current hospital examination.
In the Limitations section, we added the following:
Line 560: “We did not use any structured measurement of intelligence test to assess the mental ability of children included in our study to reduce the study-load for the included children. Instead, children were encouraged to indicate if they could understand the questions at the start of the study. Furthermore, each child was accompanied by a study mentor ( i.e. parent, researcher) during the completion of the self-rating questionnaire which made it possible for children to ask for information if needed. ”
“Q4. I was also surprised that the diagnostic interview they used was based on DSM-IV criteria for mental disorders when the DSM-5 was published 8 years ago. Psychometric properties for the measures in the Method section were also not described accurately at times (e.g., confusing validity and reliability) or adequately.”
We thank the reviewer for this feedback. We’d like to take this opportunity to clarify why we have applied DSM-IV instead of DSM-5 for this study. The main reason for this is that it has unfortunately taken several years until the english version of the MINI Kid was updated to DSM-5. After that, it took some additional time until the Hungarian version of the MINI Kid was adapted to it (the last author of this paper lead the adaptation). When the data collection of the present study began, the updated MINI Kid was not completed yet. Currently, of course, our research group uses the DSM-5 version of Mini Kid in all studies.
We do believe that the changes between the two versions of the DSM are not essentially in the case of the diagnoses of ADHD in children (Epstein and Loren, 2013). The symptom criteria have not changed, only the descriptions to facilitate the adult diagnosis (Epstein and Loren, 2013). While the number of necessary ADHD symptoms above age 17 has changed from six to five and the onset of symptoms and impairments changed from 7 to 12, according to recent studies these changes have not affected childhood prevalence (Thomas, Sanders, Doust, Beller, Glasziou, 2015)
In the revised manuscript, we have added the following paragraph to the limitations of the Discussion section:
Line 565: “It is considered a limitation of the study that the MINI Kid diagnostic interview which was used in the study was still based on DSM-IV criteria instead of DSM-5. The reason for this was that the DSM-5 based version of the MINI Kid was not yet available at the start of the present study. However, we believe that the differences between the two versions are not essentially in the case of the diagnoses of ADHD in children (56). While the number of necessary ADHD symptoms above age 17 has changed from six to five and the onset of symptoms and impairments changed from 7 to 12, according to recent studies these changes have not affected childhood prevalence (57).
We also appreciate the reviewer's feedback in regard to the psychometric properties for the measures. To improve the readability of the Methods section, we omitted the psychometric data that referred to the results of previous studies, while we kept the psychometric data measured in the present study sample. The relevant table has been revised as follows (line 307):
Table 1. Descriptive statistics and internal consistency of the measurements in the present sample
|
|
Range |
M (SD) |
Median |
IQR |
Cronbach’s α |
|
ILK - Judgement by parents |
12-35 |
28.12 (5.40) |
28 |
10 |
.911 |
|
ILK - Judgement by children |
17-35 |
29.17 (3.85) |
30 |
6 |
.682 |
|
SDQ - Peer relationship problems |
0-9 |
2.47 (2.43) |
2 |
4.50 |
.788 |
|
SDQ - Prosocial behaviour |
0-10 |
7.43 (2.29) |
8 |
3.50 |
.814 |
“Q5.The Results were well organized and generally clear (with the exception of typos and English language usage errors). However, no power analysis was presented to determine if it even made sense to conduct analyses comparing the ADHD only group with the ADHD + comorbid externalizing disorders group, given the small sample sizes. Again, sample size was mentioned as a limitation, but a lack of power was not adequately discussed as a possible explanation for the findings.”
Based on the reviewer’s suggestion, a post-hoc power analysis was calculated by G*Power software to decide wheather the present sample size was large enough to detect significant differences. Based on this power analysis results, we have added the following paragraphs to the Results and the Discussion section:
Line: 372: “Post hoc power analysis was calculated using G*Power software for detecting significant effects of the results analysed (51).”
Line 382, 383, 448, 451:
“3. Results
H1. Prosocial behaviour and peer relationship problems in the treatment naïve ADHD and control groups
The treatment naïve ADHD group showed a significantly lower value in prosocial behaviour than the control group (power * .77).The treatment naïve ADHD group’s value in peer relationship problems was significantly higher than that of the control group (power * 1.00) (see Table 4).
H5. Comorbid externalising problems, prosocial behaviour and peer relationship problems in the treatment naïve ADHD group.
The two treatment naïve ADHD groups—i.e., 1) children diagnosed with externalising comorbid disorders (CD and/or ODD); and 2) children not diagnosed with externalising comorbid disorders— showed no significant difference in prosocial behaviour (power * .37). The two groups showed a significant difference in peer relationship problems (power * .81). The ADHD group with externalising comorbid disorders showed higher values in peer relationship problems than the ADHD group without externalising disorders (see Table 6).”
Line 545: “We have to mention that the statistical power was low in testing this hypothesis due to a relatively small sample size.
Line 572: Comparison of prosocial behavior in children with ADHD and ADHD with comorbid externalising disorders would need to be tested on a larger sample since in this case the power value was 0.38. For the other hypotheses, the statistical power was adequate (H1/1: power * .77; H1/2: power * 1.00; H5/2: power * .81) .”
“Q6.The Discussion was well organized and provided the reader with some good information about why the study was important in terms of using the research to identify potential targets for intervention. However, I thought the researchers could have provided more of an explanation for the different pattern of relationships when examining parent-reported QoL compared to child-reported QoL. “
Based on the reviewer's feedback, we have added the following additional information in the Discussion section, in regard to the differences between proxy, i.e. parents' and self, i.e. children's quality of life judgments:
Line: 492: “When examining the quality of life of children, it is increasingly clear that in addition to the child's own subjective judgment, an external observer is also important. The use of proxy reports, mostly the parent - is recommended in order to get a more complete and reliable picture of the situation of children and adolescents (52). According to Mattejat et al. (48), the evaluation by a parental or a caregiver is both subjective and objective assessments because, although they appear as external observers, they are themselves affected by the child’s condition. According to Thaulow and Jozefiak (53), the parent-child difference in quality of life perceptions is due to children with ADHD being more likely to focus on the present aspects, while their parents are more likely to focus on the child’s future, which is concerned with school and social problems.”
“Q7. I also thought the authors could have explained the limitations of the study more including how those limitations could have impacted their results, rather than just listing the limitations.”
We would like to thank the reviewer for this feedback. In the revised manuscript, we have now added the following paragraphs to the limitation section:
Line 553: „ Based on our results, we can however state that we found significant differences between healthy children and treatment naïve children with ADHD in the studied variables, which highlights the importance to learn more about prosociality and peer relationships through further research. “
Line 560: “we did not use any structured measurement of intelligence test to assess the mental ability of children included in our study to reduce the study-load for the included children. Instead, children were encouraged to indicate if they could understand the questions at the start of the study. Furthermore, each child was accompanied by a study mentor ( i.e. parent, researcher) during the completion of the self-rating questionnaire which made it possible for children to ask for information if needed.”
Line 565: “It is considered a limitation of the study that the MINI Kid diagnostic interview which was applied in the study was still based on DSM-IV criteria instead of DSM-5. The reason for this was that the DSM-5 based version of the MINI Kid was not yet available at the start of the present study. However, we believe that the differences between the two versions are not essentially in the case of the diagnoses of ADHD in children (56). While the number of necessary ADHD symptoms above age 17 has changed from six to five and the onset of symptoms and impairments changed from 7 to 12, according to recent studies these changes have not affected childhood prevalence (57).”
“Q8. Lastly, there was no discussion of future research directions.”
We would like to thank the reviewer for this feedback. In the revised manuscript, we have added the following paragraphs to the Discussion section:
Line 576: “Results of the present study indicate that peer relationship problems, prosocial behavior, and their relationship to quality of life of children with ADHD are important areas for future research, preferably in a longitudinal design. Understanding which factors play a role in prosociality and peer relationships in children diagnosed with ADHD, could provide valuable insights in the development of ADHD symptoms, as well as their quality of life.”
Q9. In terms of the writing, I generally appreciated the concise nature of the article. However, in addition to the typos and English language usage errors there were several instances of causal language (e.g., "effect" and "influence") being used to describe correlational findings, which should be changed.
The revised manuscript has been checked for grammar and typos by a professional in English.

Reviewer 2 Report
I appreciate the opportunity to review the manuscript “Prosocial Behaviour and Peer Relationships in Attention Deficit Hyperactivity Disorder and their Association with Comorbid Externalising Disorders and Quality of Life”. This study examined the association between prosocial behaviour, peer relationships and quality of life in treatment naïve ADHD samples compared with control healthy subjects. I have several suggestions for improvement the quality of the paper.
- The introduction is quite sparsely cited and would benefit from integration with previous literature when the authors describe the core symptoms of ADHD (Line 69) and treatment (Line 89) for ADHD population (for example, the authors could see the studies of Fabio et al. 2018; 2019; 2020 and Caprì et al., 2020).
- To state that (line 92) "there has not been any research done which explored prosocial behaviour and peer relationships among treatment naïve children with ADHD, furthermore which explored the relationship between prosocial behaviour, peer relationships and quality of life" is not a good rationale to justify the current study. Please, provide a strong rationale underlying this study and specify the hypothesis of the study.
- Please, move the "Characteristics of sample" from results to participants.
- The section results is not clear. Please, rewrote it in a single paragraph.
- Given that the authors used non-parametric analyisis, please delete Media and Standard Deviation in each table and calucate median and interquartile range (IQR) for each variables and insert them in table.
- I suggest to discuss better the implications for practice and society.
Author Response
„I appreciate the opportunity to review the manuscript “Prosocial Behaviour and Peer Relationships in Attention Deficit Hyperactivity Disorder and their Association with Comorbid Externalising Disorders and Quality of Life”. This study examined the association between prosocial behaviour, peer relationships and quality of life in treatment naïve ADHD samples compared with control healthy subjects. I have several suggestions for improvement the quality of the paper.”
Q1. „The introduction is quite sparsely cited and would benefit from integration with previous literature when the authors describe the core symptoms of ADHD (Line 69) and treatment (Line 89) for ADHD population (for example, the authors could see the studies of Fabio et al. 2018; 2019; 2020 and Caprì et al., 2020).”
We would like to thank the reviewer for this feedback. In the revised manuscript we have now added more extensive discussion of literature to the Introduction section (see below):
Line 80:“…ADHD is associated with cognitive, social and emotional impairments [20, 21] and negatively affects the child’s relationship with family members [22-24]…”
Line 121: “In the last decade several researchers have investigated the effect of ADHD on the quality of life [30-31]. Previous studies have confirmed that ADHD has a negative influence on quality of life [30-31]. Effective multimodal treatment is available for the management of ADHD, including parental education, cognitive behavioural therapy and medication [34-37].
Q2. To state that (line 92) "there has not been any research done which explored prosocial behaviour and peer relationships among treatment naïve children with ADHD, furthermore which explored the relationship between prosocial behaviour, peer relationships and quality of life" is not a good rationale to justify the current study. Please, provide a strong rationale underlying this study and specify the hypothesis of the study.
We would like to thank the reviewer for drawing our attention to the need to justify our research questions. Based on this feedback we have added the following paragraph to the Introduction section:
Line 122: “Peer relationship is important for social development of children (38). Children diagnosed with ADHD often have difficulties developing peer relationships due to their ADHD symptoms, such as impulsivity and poor attention (39). Therefore, it is important to examine possible factors connecting to their social functioning, such as their prosocial behavior. Furthermore children with ADHD have significantly lower quality of life compared to healthy children in many areas including peer relations (40). Previous studies have indicated that quality of life can be an important tool to measure the impact of a mental disorder and for assessing the effectiveness of a treatment (29).
Line 129: Based on these findings, we examined if a better understanding of peer relationship and prosocial behavior can improve the quality of life of children with ADHD. Additionally, the assessment of prosocial skills and prosocial behavior in treatment naïve children with ADHD can serve as a baseline measurement for monitoring the efficacy of therapies.”
We also formulated the following hypotheses (Line 151-163):
“Hypothesis 1. Treatment naïve children with ADHD show a lower lever of prosocial behaviour and a higher level of peer relationship problems than healthy children.
Hypothesis 2. A higher level of prosocial behaviour is associated with a higher level of quality of life in treatment naïve children with ADHD and healthy children based both on parental proxy- and children’s self-reports.
Hypothesis 3. A higher level of peer relationship problems is associated with a lower level of quality of life in the treatment naïve children with ADHD and healthy children based both on parental proxy and children’s self-report.
Hypothesis 4. A lower level of prosocial behaviour is associated with a higher level of peer relationship problems in treatment naïve children with ADHD and healthy children.
Hypothesis 5. Treatment naïve children diagnosed with ADHD and comorbid externalizing disorders show a lower level of prosocial behaviour and a higher level of peer relationship problems than treatment naïve children with ADHD and without comorbid externalizing disorders.”
Q3.Please, move the "Characteristics of sample" from results to participants.
Based on the reviewer’s suggestion „Characteristics of sample” has been moved from Results to the Participants section (Line 239-270).
Q4. The section results is not clear. Please, rewrote it in a single paragraph.
We would like to thank to the reviewer for this feedback. In the revised manuscript, we have now included all results in a single paragraph, and we have indicate to which hypothesis (H1, H2, etc.) they refer (Line 376-455).
“3. Results
H1. Prosocial behaviour and peer relationship problems in the treatment naïve ADHD and control groups
The treatment naïve ADHD group showed a significantly lower value in prosocial behaviour than the control group (power * .77).The treatment naïve ADHD group’s value in peer relationship problems was significantly higher than that of the control group (power * 1.00) (see Table 4)
H2-4. Prosocial behaviour’s association with peer relationships and quality of life.
Table 5 presents the relationship between prosocial behaviour, peer relationship problems and quality of life.
Prosocial behaviour had a weak positive relationship with the parent’s evaluation of the child’s quality of life both in the ADHD and in the control group. The parent’s rating of quality of life and peer relationship problems showed a significant, negative moderate association also in the ADHD and in the control group. Between prosocial behaviour and peer relationship problems a negative weak association was detected in the ADHD group, and a negative moderate association in the control group. Furthermore, in the ADHD group the child’s evaluation of quality of life showed a significant negative weak relationship with peer relationships, and there was no significant relationship with prosocial behaviour. In the control group the child’s view of life quality did not show a significant relationship with the other items.
H5. Comorbid externalising problems, prosocial behaviour and peer relationship problems in the treatment naïve ADHD group.
The two treatment naïve ADHD groups—i.e., 1) children diagnosed with externalising comorbid disorders (CD and/or ODD); and 2) children not diagnosed with externalising comorbid disorders— showed no significant difference in prosocial behaviour (power * .37). The two groups showed a significant difference in peer relationship problems (power * .81). The ADHD group with externalising comorbid disorders showed higher values in peer relationship problems than the ADHD group without externalising disorders (see Table 6).”
Q5. Given that the authors used non-parametric analyisis, please delete Media and Standard Deviation in each table and calucate median and interquartile range (IQR) for each variables and insert them in table.
We would like to thank the reviewer for this feedback. Mean and standard deviation data could be informative to readers, so we included them in the tables, however, the additions requested by the reviewer are listed in the Table 1, Table 4 and Table 6 as follows (Line: 307, 385, 452 )
Table 1. Descriptive statistics and internal consistency of the measurements in the present sample
|
|
Range |
M (SD) |
Median |
IQR |
Cronbach’s α |
|
ILK - Judgement by parents |
12-35 |
28.12 (5.40) |
28 |
10 |
.911 |
|
ILK - Judgement by children |
17-35 |
29.17 (3.85) |
30 |
6 |
.682 |
|
SDQ - Peer relationship problems |
0-9 |
2.47 (2.43) |
2 |
4.50 |
.788 |
|
SDQ - Prosocial behaviour |
0-10 |
7.43 (2.29) |
8 |
3.50 |
.814 |
Note. M=Mean;, SD=Standard deviation; IQR= interquartile range
Table 4. Difference between ADHD and control group in prosocial behaviour and peer relationship problems
|
|
ADHD group (n=79) |
Control group (n=54) |
|
||||
|
M (SD) |
Median |
IQR |
M(SD) |
Median |
IQR |
Mann Whitney U test |
|
|
Prosocial behaviour |
6.66(2.40) |
7 |
4 |
8.31(1.84) |
9 |
3 |
U=1114.50; p<.0001 |
|
Peer relationship problems |
4.05(2.47) |
4 |
4 |
0.73(0.90) |
0 |
1 |
U=3244.00; p<.0001 |
Note. M=Mean;, SD=Standard deviation; IQR= interquartile range
Table 6. Prosocial behaviour and peer relationship problems among children diagnosed with ADHD and among children diagnosed with ADHD and externalising comorbid diagnoses
|
|
ADHD group (n=29) |
ADHD + CD and/or ODD group (n=49) |
|
||||
|
M (SD) |
Median |
IQR |
M(SD) |
Median |
IQR |
Mann Whitney U test |
|
|
Prosocial behaviour |
7.14(2.28) |
7 |
4.75 |
6.38(2.45) |
7 |
3 |
U=539.00; p=.279 |
|
Peer relationship problems |
3.08(2.59) |
2 |
4 |
4.58(2.25) |
4.5 |
3 |
U=771.50; p=.017 |
Note. M=Mean;, SD=Standard deviation; IQR= interquartile range
Q6. I suggest to discuss better the implications for practice and society.
We would like to thank the reviewer for this feedback. In the revised manuscript, we have now added the folowing paragraph to the Conclusions part of the manuscript:
Line 638: “Social support from parents, teachers, as well as peers is important for all patients with a psychiatric disorders, including ADHD. Psychoeducation can be an important factor for parents and teachers to improve acceptance and support towards children with ADHD. However, peers cannot be expected to be more accepting and tolerant to children with ADHD as a result of psychoeducation, because they are themselves children as well. Therefore, children diagnosed with ADHD may need extra support to improve their relationship with their peers. Reducing social exclusion and improving peer relationships in addition to effective medication and non-medication therapies can help to protect children diagnosed with ADHD from future loneliness or deviations.”

Reviewer 3 Report
This is an interesting study with a good rationale for conducting the research.
I have a few minor changes to improve the readability:
- Line 149 change to: The modified version of the MINI Kid evaluates not only psychiatric disorders, but assesses all psychiatric symptoms, enabling subthreshold disorders to be identified.
- Line 152, there appears to be an apostrophe placed in error "while ’those".
- Line 164, do you mean internal consistency? "validation of the clinical population, the subscales’ inner consistency"
- Line 165 onwards change to: The psychometric characteristics of the Hungarian version of the SDQ subscales, in terms of gender, age, and evaluator, are in line with international results [40].
- Line 174/5, missing an “and” and spelling error on proxy: for children from 7 to 18 years of age. It has a self-rated version both for children and adolescents, and a proxy parent rated version as well.
- Line 176, change to internal consistency
- Line 187 “The salient cases have been excluded”. Please can you provide more details about this – I am not clear what you mean?
- Line 306 remove “on” The findings highlight on that treatment naïve children with ADHD do not experience an association…”
- Line 338 spelling error - change “Forth” to Fourth
Throughout the document you change between past and present text – being consistent throughout would improve readability.
Author Response
„This is an interesting study with a good rationale for conducting the research.
I have a few minor changes to improve the readability:
We would like to thank the reviewer for acknowledging the relevance of our topic. Based on the reviewer’s feedback we have revised the manuscript (see below).
- Line 149 change to: The modified version of the MINI Kid evaluates not only psychiatric disorders, but assesses all psychiatric symptoms, enabling subthreshold disorders to be identified.
The sentence has been revised as suggested (line 283).
- Line 152, there appears to be an apostrophe placed in error "while ’those".
The sentence has been revised as suggested (line 286).
- Line 164, do you mean internal consistency? "validation of the clinical population, the subscales’ inner consistency"
We would like to thank the reviewer for the feedback. However, based on the feedback of one of the other reviewers, this part has been deleted in the revised manuscript.
- Line 165 onwards change to: The psychometric characteristics of the Hungarian version of the SDQ subscales, in terms of gender, age, and evaluator, are in line with international results [40].
We would like to thank the reviewer for the feedback. However, based on the feedback of one of the other reviewers, this part has been deleted in the revised manuscript.
- Line 174/5, missing an “and” and spelling error on proxy: for children from 7 to 18 years of age. It has a self-rated version both for children and adolescents, and a proxy parent rated version as well.
The sentence has been revised as suggested (line 304).
- Line 176, change to internal consistency
We would like to thank the reviewer for the feedback. However, based on the feedback of one of the other reviewers, this part has been deleted in the revised manuscript.
- Line 187 “The salient cases have been excluded”. Please can you provide more details about this – I am not clear what you mean?
We thank the reviewer for this feedback, we have revised this part as follows:
Line 362: „Salient cases have been excluded, i.e. cases where the response exceeded the minimum or maximum score of the questionnaire.”
- Line 306 remove “on” The findings highlight on that treatment naïve children with ADHD do not experience an association…”
In the revised manuscript, this sentence has been removed (line 516).
- Line 338 spelling error - change “Forth” to Fourth
In the revised manuscript manuscript, this has been changed (line 565).
Throughout the document you change between past and present text – being consistent throughout would improve readability.
The revised manuscript has checked by a proof reader.

Round 2
Reviewer 1 Report
Authors did a fine job addressing comments.
This manuscript is a resubmission of an earlier submission. The following is a list of the peer review reports and author responses from that submission.
Round 1
Reviewer 1 Report
This manuscript describes a study in which the researchers examine whether the Attention Deficit Hyperactivity Disorder (ADHD) has a negative influence on the child’s peer relationships and quality of life. The authors report that prosocial behaviour has a weak positive relationship with parents’ rating of the child’s quality of life. I am not sure, however, that this paper makes a significant contribution to the literature in its present form. The research question is not well situated within the literature, it's not clear what this paper is bringing to the table. There are serious problems with the approach that prevent the study from addressing that question at the very high level of rigor and quality required for publication.
Abstact: no number, age and gender of participants
Introduction: unclear rationale underlying this study
Analysis and results: the effects (Fisher) are not adequately explained. No test t or post-hoc analysis.
Discussion: this seemed very limited/basic in interpretation.
These issues appear to be endemic to the design of the study.
Reviewer 2 Report
Although research on quality of life of children, adolescents and adults with ADHD are still important for the disorder-related risk prevention, the article doesn’t bring any important novelty in this topic in my opinion. The relations revealed in the study seems to be obvious and well presented in the previous research. Unfortunately, the study quality is also affected by small size of the sample and control group, as well as the sex proportions. Specific comments for authors: Abstract: “ Prosocial behaviour has a weak positive relationship with parents’ rating of the child’s quality of life”. “According to the child’s rating, quality of life showed a significant negative weak relationship with peer relationships in the ADHD group”. Is this in line with the data presented in the results (particularly in table 3)? Materials and Methods How the group of treatment naive children was defined? They were recruited from hospital and outpatient clinic. I assume, that in such institutions it is hard to find children who have never been treated eg.: with stimulants. Do you mean only pharmacological treatment or also psychotherapeutic interventions? What was the criterion for mental retardation? Authors should describe randomisation procedure. “Exclusion criteria in the control group were mental retardation in the medical history…” Does it mean, that researcher had access to the medical records? Please, explain. The number of Ethical Commitee decission should be mentioned “children older than 14 years of age” - I suggest using the word "adolescents" instead of children in case of individuals older than 12y. The definition/ criterion for ADHD, CD, ODD should be described in details. What modification was applied to the original MINI Kid? “First the questionnaire was filled out by non-clinical 6 year-old children’s parents” - Why? Could you explain this part of procedure and why this specific age? Results: Is it possible to provide an attrition analysis? How many children were excluded from the control group because of psychiatric disorders and mental retardation? How many children from the sample were excluded because of mental retardation or because were treated pharmacologically What were other than ODD and CD comorbid disorders? What was a proportion of boys and girls in groups with and without comorbid disorders? Discussion Authors didn't address the negative relation between QoL reported by parents and prosocial behaviour in controls.